# High Rates of Sexualized Drug Use or Chemsex among Brazilian Transgender Women and Young Sexual and Gender Minorities

**DOI:** 10.3390/ijerph19031704

**Published:** 2022-02-02

**Authors:** Emilia M. Jalil, Thiago S. Torres, Claudia C. de A. Pereira, Alessandro Farias, Jose D. U. Brito, Marcus Lacerda, Daila A. R. da Silva, Nickols Wallys, Gabriela Ribeiro, Joyce Gomes, Thiffany Odara, Ludymilla Santiago, Sophie Nouveau, Marcos Benedetti, Cristina Pimenta, Brenda Hoagland, Beatriz Grinsztejn, Valdilea G. Veloso

**Affiliations:** 1Instituto Nacional de Infectologia Evandro Chagas, Fundação Oswaldo Cruz (INI-Fiocruz), Rio de Janeiro 21040-360, Brazil; thiago.torres@ini.fiocruz.br (T.S.T.); wallywae@gmail.com (N.W.); gabi_csirj@hotmail.com (G.R.); marcos.benedetti@ini.fiocruz.br (M.B.); brenda.hoagland@ini.fiocruz.br (B.H.); gbeatriz@ini.fiocruz.br (B.G.); valdilea.veloso@ini.fiocruz.br (V.G.V.); 2Escola Nacional de Saúde Pública Sérgio Arouca ENSP-Fiocruz, Rio de Janeiro 21041-210, Brazil; pereirac.claudia@gmail.com; 3Centro Estadual Especializado em Diagnóstico, Assistência e Pesquisa (CEDAP), Salvador 40100-010, Brazil; farias.alessandro@gmail.com (A.F.); thiffany.odara@gmail.com (T.O.); 4Hospital Dia Asa Sul, Brasília 70351-580, Brazil; jdub.jdub15@gmail.com (J.D.U.B.); santtiagobsb@gmail.com (L.S.); 5Fundação de Medicina Tropical Doutor Heitor Vieira Dourado (FMT), Manaus 69040-000, Brazil; marcuslacerda.br@gmail.com (M.L.); joycegomesfb@gmail.com (J.G.); 6Centro de Testagem Aconselhamento (CTA) Santa Marta, Secretaria Municipal de Saúde de Porto Alegre, Porto Alegre 90010-040, Brazil; daila@portoalegre.rs.gov.br (D.A.R.d.S.); sophienouveauw@gmail.com (S.N.); 7Departamento de Condições Crônicas e Infecções Sexualmente Transmissíveis, Secretaria de Vigilância em Saúde, Ministério da Saúde, Brasília 70719-040, Brazil; cristina.pimenta@aids.gov.br

**Keywords:** sexual and gender minorities, chemsex, transgender women, men who have sex with men, Latin America, Brazil

## Abstract

(1) Background: We aimed to estimate sexualized drug use (SDU) prevalence and its predictors among sexual and gender minorities. (2) Methods: We used an online and on-site survey to enroll sexual/gender minorities people between October–December/2020, and multivariate logistic regression to obtain SDU correlates. (3) Results: We enrolled 3924 individuals (280 transgender women [TGW], 3553 men who have sex with men [MSM], and 91 non-binary), 29.0% currently on pre-exposure prophylaxis (PrEP). SDU prevalence was 28.8% (95% confidence interval [CI] 27.4–30.2). TGW had 2.44-times increased odds (95%CI 1.75–3.39) of engaging in SDU compared to MSM, regardless of PrEP use. PrEP use (aOR 1.19, 95%CI 1.00–1.41), South/Southeast region (aOR 1.26, 95%CI 1.04–1.53), younger age (18–24 years: aOR 1.41, 95%CI 1.10–1.81; 25–35 years: aOR 1.24, 95%CI 1.04–1.53), white race/color (aOR 1.21, 95%CI 1.02–1.42), high income (aOR 1.32, 95%CI 1.05–1.67), binge drinking (aOR 2.66, 95%CI 2.25–3.14), >5 sexual partners (aOR 1.88, 95%CI 1.61–2.21), condomless anal sex (aOR 1.49, 95%CI 1.25–1.79), self-reported sexually transmitted infection (aOR 1.40, 95%CI 1.14–1.71), and higher perceived HIV-risk (aOR 1.37, 95%CI 1.14–1.64) were associated with SDU. (4) Conclusions: TGW had the highest SDU odds. SDU may impact HIV vulnerability among key populations and should be addressed in HIV prevention approaches.

## 1. Introduction

Sexualized drug use (SDU) is the intentional use of illicit drugs before and/or during sex to sustain, enhance, disinhibit, or facilitate the sexual experience [1]. Often ‘chemsex’, a subset of SDU, refers to the use of methamphetamines, gamma-hydroxybutyric acid/gamma-butyrolactone (GHB/GBL), or mephedrone [2], although these definitions are not consensual. Although SDU rates vary greatly depending on the country/region, population, and definition used, current evidence indicates that SDU is on the rise, particularly in high-income countries [2,3,4,5]. Nevertheless, low- and middle-income countries’ (LMIC) reports suggest an important role of SDU in the HIV epidemic, especially among key populations [6,7,8].

HIV infection disproportionately affects key populations worldwide, including in Latin America [9]. In Brazil, despite a 0.4% stable prevalence in the overall population [10], estimates among gay, bisexual and other men who have sex with men (MSM) and transgender women (TGW) reach 18.4% and 31.2%, respectively [11,12,13]. National data for Brazil point to rising incidences among young MSM aged 18–24 years [14]. One in four Brazilian TGW were already living with HIV by the age of 24 [15]. Pre-exposure prophylaxis (PrEP) is an efficacious biomedical strategy to prevent HIV among individuals at risk, including MSM and TGW [16]. Brazil started PrEP implementation in 2017, and this strategy is currently recommended for key populations engaging in risky sexual behavior [17].

SDU contributes to increasing HIV risk as it relates to increased risky sexual behaviors, such as condomless anal sex [18]. In addition, SDU is associated with increased rates of sexually transmitted infections (STI) [19,20]. Moreover, SDU is more common among MSM compared to heterosexual cisgender men and women [21,22,23,24,25]. Although SDU could be perceived as a barrier for PrEP use, some data indicate that individuals reporting SDU had increased rates of post-exposure prophylaxis (PEP) and PrEP use [4]. In addition, recent data showed no evidence that SDU interfered with PrEP persistence [26].

There is a dearth of data on SDU among key populations in Latin America. A few Latin-American studies that addressed SDU among MSM reported prevalences ranging from 4–36.6% [6,27]. There is no data on SDU among Latin-American TGW and non-binary individuals. As PrEP has been slowly incorporated as an HIV prevention strategy in Latin America, it is urgent to gather data on SDU within key populations assessing PrEP. To address this gap, we aimed to estimate SDU prevalence and its predictors among sexual and gender minorities in Brazil.

## 2. Materials and Methods

### 2.1. Study Design

This was a cross-sectional study using on-site and online strategies to recruit sexual and gender minorities to participate in a discrete choice experiment (DCE) from October to December 2020. Details of the study design are described elsewhere [28]. Briefly, inclusion criteria were (i) being aged 18+ years, (ii) being cisgender MSM, TGW or non-binary, and (iii) self-reporting as HIV-negative. On-site recruitment occurred in five PrEP/HIV services in all country regions: North (Manaus), Northeast (Salvador), Central-west (Brasília), Southeast (Rio de Janeiro) and South (Porto Alegre). Online recruitment occurred through advertisements on geosocial networking (GSN) and apps (Hornet and Grindr) targeting sexual and gender minorities living in any Brazilian city.

### 2.2. Outcome

SDU or ‘chemsex’ was assessed using the question: “In the last 6 months, have you used any illicit drug before/during sex?”

### 2.3. Primary Exposure Variable

Individuals were asked about PrEP use as a multiple choice question with possible answers: never, current and past.

### 2.4. Covariables

We collected sociodemographic characteristics such as gender (cisgender men, TGW, non-binary or gender fluid), country region (North, Northeast, Central-west, Southeast and South), age (collected continuous; presented in median/interquartile range [IQR] and categorized in 18–24, 25–35, >35 years), race/skin color (Asian, Black, Indigenous, Pardo or Mixed-Black and White; due to the low sample size Asian (N = 46) was grouped with White and Indigenous (N = 26) with Pardo [29]), completed schooling (elementary, secondary and post-secondary), family monthly income (low: <USD 400, middle: USD 401–1200, high: >USD 1200) and sexual orientation (gay/homosexual and other).

We asked about substance use (pre-existing list with an additional open answer) in the past 6 months. We report any illicit drug use (yes/no), tobacco use, marijuana use and stimulant drug use (cocaine, crack, ecstasy or other amphetamines, ketamine, meth, GHB, poppers or other inhalants). Binge drinking was evaluated as: “In the last six months, did you drink five or more drinks in a couple of hours?”. A dose was defined as 1 can of beer (300 mL) or 1 glass of wine (120 mL) or 1 shot of distilled alcohol (30 mL of ex. *cachaça*, vodka, whisky, tequila, mezcal, or pisco) [30].

We also assessed sexual behavior in the previous 6 months by number of sexual partners (collected continuous, provided as median/IQR and dichotomized in ≤5 and >5), condomless anal sex (yes/no), condomless receptive anal sex (yes/no), steady partner (yes/no), and sexually transmitted infection (STI) (yes/no). HIV perceived risk was assessed with the question: “What is your chance of getting HIV in the next year?”, with possible answers: no, low, moderate, high, 100% sure and I don’t want to answer/I don’t know (considered missing for this analysis; N = 175, 4.5%). Due to the low sample size of some categories and following previous studies [29,31], we further dichotomized into: no/low (779 [19.9%] and 2103 [53.6%], respectively) and moderate/high/100% sure (680 [17.3%], 173 [4.4%] and 14 [0.3%], respectively).

Participants answered about PrEP awareness (“Before today, have you ever heard about PrEP?”, yes/no). We defined PrEP eligibility according to the Brazilian Ministry of Health criteria [17]. For those who never used PrEP before, we inquired about willingness to use PrEP (“Would you be willing to use PrEP for HIV prevention?”) with a 5-point Likert scale; PrEP willingness was defined as ‘highly likely’ [32,33]. For those currently on PrEP, we assessed their PrEP modality (daily oral PrEP, event-driven [ED]-PrEP or injectable PrEP [cabotegravir]). For those reporting daily oral PrEP use, we inquired about complete adherence (no missing pills) during the past 30 days and 7 days. Lastly, we asked about PEP use in the past 12 months.

### 2.5. Statistical Methods

First, we described the study population according to current PrEP use. We also compared the variables by gender (cisgender men, TGW and non-binary) using chi-square or Ranksum tests for these comparisons, and evaluated effect size using Cramer’s V coefficient. We conducted a bivariate analysis to explore the unadjusted association of the outcome (SDU) with the primary exposure factor (current PrEP use) and other variables, such as sociodemographic, sexual behavior and HIV perceived risk. Lastly, we performed logistic regression models to identify factors associated with SDU overall and according to PrEP use; our hypothesis is that these associations may differ among individuals on and off-PrEP, as previously described [4]. To obtain the multivariable model for SDU overall, we used a backward stepwise modeling approach; variables with a bivariate *p*-value < 0.02 were included in the initial model and subsequently excluded if *p*-value > 0.05. Gender, geographic region, race/color, completed schooling, and family monthly income were defined as confounders a priori and maintained in the final model regardless of statistical significance. Exclusions occurred by each variable, beginning with the variable with the highest *p*-value. After each exclusion, we ran a new model before repeating the process. After this process, the final multivariable model included variables that remained significant (*p*-value ≤ 0.05) and those defined as confounders a priori. Multivariable models according to PrEP use were obtained with the same process but included the same variables retained in the final multivariable model for SDU overall besides those remaining significant. We tested for multicollinearity of variables retained in final multivariable models by using the variance inflation factor (VIF). All analyses were performed using R version 4.0.5 (The R project).

## 3. Results

### 3.1. Study Population

We enrolled 3924 sexual and gender minorities people (280 [7.1%] TGW, 3553 [90.5%] MSM, and 91 [2.3%] non-binary), 29.0% of them currently on PrEP (n = 1139). PrEP use did not differ according to gender identity (Table 1). TGW had lower PrEP awareness, lower current PrEP use, and higher PrEP willingness compared to MSM and non-binary participants (Appendix A). TGW were more frequently recruited onsite (275/280, 98.2% onsite vs. 5/280, 1.8% online compared to MSM (679/3553, 19.1% onsite vs. 2874/3553, 80.9% online). Number of non-binary individuals recruited onsite and online was similar (47/91, 51.6% vs. 44/91, 48.3%).

Individuals off-PrEP were more commonly recruited online (81.5%) than onsite (18.5%). PrEP users were older, had higher schooling, higher prevalence of illicit drug use, including stimulants, higher number of sexual partners, more frequently engaged in condomless anal sex, and more frequently reported STIs compared to individuals off-PrEP. No or low HIV perceived risk was more common among PrEP users than those off-PrEP (88.1% vs. 72.2%). Among participants on PrEP (n = 1139), 92.5% were on daily oral PrEP (1048), 2.9% (33.0) reported ED-PrEP, and 4.6% (52) were on injectable PrEP. Both injectable PrEP and ED-PrEP were more common among non-binary participants (0 = 0.006) (Appendix A).

### 3.2. Prevalence of SDU and Associated Factors among Sexual and Gender Minorities

SDU prevalence was 28.8% (95% confidence interval [CI] 27.4–30.2). Table 2 describes the characteristics of the study population stratified by SDU overall and according to PrEP use. The SDU rates were highest among TGW (40.7%, 95%CI 35.0–46.7), followed by non-binary participants (33.0%, 95%CI 23.7–43.7), and MSM (27.8%, 95%CI 26.3–29.3), and did not differ according to recruitment. TGW had 2.44-times increased odds (95%CI 1.75–3.39) of engaging in SDU compared to MSM, regardless of PrEP use (aOR 2.11, 95%CI 1.13–3.92, among TGW on PrEP and aOR 2.54, 95%CI 1.71–3.77 among TGW off-PrEP) (Table 3).

Overall, in addition to TGW identity, PrEP use (aOR 1.19, 95%CI 1.00–1.41), South/Southeast country region (aOR 1.26, 95%CI 1.04–1.53), younger age (aOR 1.41, 95%CI 1.10–1.81 for 18–24 years, and aOR 1.24, 95%CI 1.04–1.53 for 25–35 years), white race/color (aOR 1.21, 95%CI 1.02–1.42), high income (aOR 1.32, 95%CI 1.05–1.67), binge drinking (aOR 2.66, 95%CI 2.25–3.14), more than 5 sexual partners (aOR 1.88, 95%CI 1.61–2.21), condomless anal sex (aOR 1.49, 95%CI 1.25–1.79), self-reported STI (aOR 1.40, 95%CI 1.14–1.71), and higher perceived HIV-risk (aOR 1.37, 95%CI 1.14–1.64) were associated with SDU.

Among PrEP users, in addition to TGW identity, SDU predictors were: South/Southeast country region [aOR 1.64, 95%CI 1.16–2.34), age between 25–35 years (aOR 1.54, 95%CI 1.12–2.13), binge drinking (aOR 2.85, 95%CI 2.12–3.86), >5 sexual partners (aOR 1.93, 95%CI 1.45–2.57) and self-reported STI (aOR 1.53, 95%CI 1.11–2.09). Among participants off-PrEP, in addition to TGW, non-binary (aOR 1.85, 95%CI 1.04–3.21) participants had increased odds of SDU as compared to MSM. Individuals off-PrEP aged 18–24 years (aOR 1.42, 95%CI 1.06–1.89), reporting binge drinking (aOR 2.56, 95%CI 2.10–3.14), >5 sexual partners (aOR 1.86, 95%CI 1.53–2.25), condomless anal sex (aOR 1.63, 95%CI 1.33–2.01) and higher HIV perceived risk (aOR 1.38, 95%CI 1.13–1.70) also had higher odds of SDU.

## 4. Discussion

Almost a third of the individuals enrolled in this analysis reported SDU, with higher rates for TGW and non-binary individuals. The multivariable analysis observed that irrespective of PrEP use, SDU was more prevalent among TGW, the population most impacted by HIV infection in Brazil [12]. Furthermore, young individuals and those on PrEP had increased odds of SDU. Behavioral variables, such as binge drinking and risky sexual behavior, were independently associated with SDU. To our knowledge, this is the largest Latin-American sample to evaluate SDU in the context of PrEP use and the first to report SDU according to gender identity in Brazil.

Although SDU estimates are difficult to compare due to the heterogeneity between studies and populations [4], our findings are higher than previous reports enrolling MSM from high income [3,4]. In general, SDU prevalences are higher in samples obtained at sexual health clinics [4], such as part of ours. A Canadian study identified that 24% of 2923 people attending a health unit for PrEP reported SDU in the last year [26]. Another Brazilian study that included 1048 participants identified a 36.6% SDU estimate [6]. Previous analyses conducted by our group found strong associations of sexual behavior and SDU among MSM from Rio de Janeiro [6].

In our analysis, TGW had more than 2-times the odds of reporting SDU as compared to MSM. SDU prevalence among TGW in this study was similar to that of another Brazilian study among *travestis* (a traditional transgender identity in Brazil that predates queer studies. This identity has been associated with sex work in the past. Although highly stigmatized, the expression has gained importance in LGBTQIA+ activism in the country.) (43%) [34]. TGW are a highly marginalized group in Brazil with high levels of structural transphobia, homicides, violence, sex work, barriers to access fundamental human rights, such as health, education and formal jobs [35]. Brazil ranks first in trans homicides worldwide [36]. Moreover, TGW bear a high burden of HIV worldwide with 49-times higher odds of HIV infection compared to other population groups [37]. Brazilian TGW have HIV prevalences as high as 30% [12,13] and extremely high HIV incidence [38]. Data from ImPrEP, an ongoing demonstration study to assess preparedness for the rollout of effective HIV prevention among key populations in Brazil, Peru and Mexico, demonstrated that TGW presented worse outcomes than MSM in the three countries [39]. Finally, the Brazilian National Surveillance System indicates that PrEP discontinuation occurs among 50% of all TGW on PrEP, in contrast to 38% among MSM [40]. SDU has been associated with condomless anal sex among TGW and may contribute to their increased vulnerability to HIV [41,42].

Age was also associated with SDU, with younger groups (18–24 and 25–35 years) showing higher odds of SDU. In contrast, most studies from developed countries reported higher median ages of people engaging on SDU ranging from 32 to 42 years [4]. Gender and sexual minorities youth may be the populational group at highest HIV risk in Brazil. Brazilian National Data showed an increase in HIV infection among cisgender men aged 15–24 years probably due to the impact of the epidemic among young MSM [43]. In addition, HIV prevalence among MSM was higher in a 2016 Brazilian survey compared to 2009 due to a higher percentage of young participants in the most recent study [44]. Individuals aged 18–24 years had a higher risk of HIV seroconversion in the ImPrEP study [45]. Among TGW, a previous study observed that trans youths had higher odds of using substances and engaging in condomless anal sex, and 24.5% of the participants aged 18–24 years were living with HIV [15].

Binge drinking was also associated with SDU. This is in agreement with a previous Brazilian study that showed a strong association between moderate/high risk for alcohol disorders and SDU [6]. As a licit drug, the risks associated with drinking alcohol have been minimized in our society. Nevertheless, alcohol use has been implicated in several sexual risky behaviors [46], some of which have been consistently associated with SDU in our analysis. In addition, we observed that PrEP users had higher odds of SDU, as previously described [4]. This uncovers the multiple and complex layers that put individuals at risk for HIV and need to be addressed to prevent HIV transmission at the population level.

This study has several limitations. First, this was a cross-sectional study, which hinders causal inferences. As such, we cannot affirm that SDU is associated with worse PrEP outcomes for instance, even though its high prevalence among most vulnerable groups, including young people and TGW, is disturbing. In addition, ours is a convenience sample. As such, our results may not be generalizable to all MSM, TGW, and non-binary persons. Nevertheless, as we used complimentary recruitment strategies, we were able to reach a large and diverse population [47]. Our sample sizes according to gender and sexual minorities were very different, which may interfere with the comparison between the groups. Furthermore, SDU prevalence may be underestimated due to social desirability bias both in in-person and web-based interviews. Most PrEP users were recruited on site and responded to face-to-face interviews, which may have interfered with our results. We only assessed family income, which may be influenced by the number of people included in the family. Additionally, we assessed sexual behaviors and drug use as dichotomic variables, with no evaluation of the different levels of drug use. We did not examine whether any of the predictors acted as moderator variables in this analysis. Finally, the study occurred during the COVID-19 pandemic, the study population and their sexual behaviors may be different from the pre-COVID-19 era. Despite these limitations, this is a large study that evaluated SDU, an understudied topic in LMIC, in view of HIV prevention disaggregated by gender and sexual minorities.

## 5. Conclusions

Although SDU was common among sexual and gender minorities, TGW had the highest SDU odds. SDU may impact HIV vulnerability among key populations, including TGW, and should be addressed in HIV prevention approaches.

## Figures and Tables

**Table 1 ijerph-19-01704-t001:** Sociodemographic characteristics, substance use, and sexual behavior of the study population according to current PrEP use.

	Current PrEP Use		Cramer’s V Coefficient ^3^	*p*-Value
	No	Yes		
	N = 2785	N = 1139		
**Gender**			0.01	0.834
Cisgender men	2521 (90.5)	1032 (90.6)		
Transgender women	197 (7.1)	83 (7.3)		
Non binary	67 (2.4)	24 (2.1)		
**Recruitment**			0.25	<0.001
On site	516 (18.5)	485 (42.6)		
Online	2269 (81.5)	654 (57.4)		
**Geographic region**			0.19	<0.001
North	55 (2.0)	108 (9.5)		
Northeast	326 (11.7)	73 (6.4)		
Central-west	247 (8.9)	78 (6.8)		
Southeast	1825 (65.5)	775 (68)		
South	332 (11.9)	105 (9.2)		
**Age**				
Median (IQR)	31 (26,37)	32 (27,38)		0.001
18–24	505 (18.1)	165 (14.5)	0.053	0.004
25–35	1420 (51.0)	574 (50.4)		
>35	860 (30.9)	400 (35.1)		
**Race/color**			0.006	0.932
White/Asian	1539 (56.0)	624 (55.6)		
Pardo/Indigenous	767 (27.9)	313 (27.9)		
Black	442 (16.1)	186 (16.6)		
**Completed schooling**			0.042	0.035
Elementary	179 (6.5)	50 (4.4)		
Secondary	782 (28.2)	341 (30.0)		
Post-secondary	1812 (65.3)	744 (65.6)		
**Family monthly income**			0.017	0.566
Low	916 (34.2)	351 (32.4)		
Middle	1085 (40.5)	453 (41.8)		
High	679 (25.3)	280 (25.8)		
**Sexual Orientation**			0.031	0.053
Gay or homosexual	2185 (78.5)	925 (81.2)		
Other	600 (21.5)	214 (18.8)		
**Any illicit drug use ^1^**	1118 (40.1)	533 (46.8)	0.061	<0.001
**Tobacco use ^1^**	903 (32.4)	351 (30.8)	0.016	0.327
**Marijuana use ^1^**	884 (31.7)	405 (35.6)	0.037	0.021
**Stimulant drug use ^1^**	591 (21.2)	323 (28.4)	0.077	<0.001
**Binge drinking ^1^**	1603 (57.6)	677 (59.4)		0.279
**Number of sexual partners ^1^**				
Median (IQR)	4 (2,10)	7 (3,15)		<0.001
≤5	1828 (65.6)	535 (47)	0.173	<0.001
>5	957 (34.4)	604 (53.0)	0.017	
**Condomless anal sex ^1^**	1765 (63.4)	923 (81.0)	0.173	<0.001
**Condomless receptive anal sex ^1^**	1230 (44.2)	718 (63.0)	0.171	<0.001
**Steady partner ^1^**	960 (34.5)	491 (43.1)	0.081	<0.001
**Reported STI ^1^**	370 (13.3)	265 (23.3)	0.123	<0.001
**HIV perceived risk ^2^ (N = 3749)**			0.171	<0.001
No/Low	1909 (72.2)	973 (88.1)		
Moderate/High/100% sure	735 (27.8)	132 (11.9)		

All results are shown as number of respondents in that category followed by percentage in parentheses; ^1^ Previous 6 months before study visit; ^2^ In the next year after the study visit; ^3^ Cramer’s V coefficients indicate small effect size for all evaluated variables.

**Table 2 ijerph-19-01704-t002:** Characteristics of the study population stratified by SDU overall and according to PrEP use.

	Overall Sample (N = 3924)		Current PrEP Use			
			No (N = 2785)		Yes (N = 1139)	
**Sexualized drug use (SDU)**	**No** **2794 (71.2)**	**Yes** **1130 (28.8)**	**No** **2039 (73.2)**	**Yes** **746 (26.8)**	**No** **755 (66.3)**	**Yes** **384 (33.7)**
**Current PrEP use**						
No	**2039 (73.2)**	**746 (26.8)**	NA	NA	NA	NA
Yes	**755 (66.3)**	**384 (33.7)**	NA	NA	NA	NA
**Gender**						
Cisgender men	**2567 (72.2)**	**986 (27.8)**	**1880 (74.6)**	**641 (25.4)**	**687 (66.6)**	**345 (33.4)**
Transgender women	**166 (59.3)**	**114 (40.7)**	**115 (58.4)**	**82 (41.6)**	**51 (61.4)**	**32 (38.6)**
Non-binary	**61 (67)**	**30 (33)**	**44 (65.7)**	**23 (34.3)**	**17 (70.8)**	**7 (29.2)**
**Geographical region**						
North/Northeast/Central-west	652 (73.5)	235 (26.5)	459 (73.1)	169 (26.9)	**193 (74.5)**	**66 (25.5)**
South/Southeast	2142 (70.5)	895 (29.5)	1580 (73.2)	577 (26.8)	**562 (63.9)**	**318 (36.1)**
**Age (years)**						
18–24	**461 (68.8)**	**209 (31.2)**	**348 (68.9)**	**157 (31.1)**	**113 (68.5)**	**52 (31.5)**
25–35	**1386 (69.5)**	**608 (30.5)**	**1028 (72.4)**	**392 (27.6)**	**358 (62.4)**	**216 (37.6)**
>35	**947 (75.2)**	**313 (24.8)**	**663 (77.1)**	**197 (22.9)**	**284 (71.0)**	**116 (29.0)**
**Race**						
White/Asian	1532 (55.5)	631 (56.7)	906 (73.1)	334 (26.9)	**356 (69.3)**	**158 (30.7)**
Black/Pardo/Indigenous	1226 (44.5)	482 (43.3)	1107 (73.4)	401 (26.6)	**389 (63.9)**	**220 (36.1)**
**Completed schooling**						
Elementary	152 (66.4)	77 (33.6)	**117 (65.4)**	**62 (34.6)**	35 (70)	15 (30)
Secondary	797 (71)	326 (29)	**565 (72.3)**	**217 (27.7)**	232 (68)	109 (32)
Post-secondary	1835 (71.8)	721 (28.2)	**1351 (74.6)**	**461 (25.4)**	484 (65.1)	260 (34.9)
**Family monthly income**						
Low	900 (71)	367 (29)	660 (72.1)	256 (27.9)	240 (68.4)	111 (31.6)
Middle	1120 (72.8)	418 (27.2)	812 (74.8)	273 (25.2)	308 (68)	145 (32.0)
High	663 (69.1)	296 (30.9)	489 (72)	190 (28)	174 (62.1)	106 (37.9)
**Binge drinking**						
No	**1354 (82.4)**	**290 (17.6)**	**990 (83.8)**	**192 (16.2)**	**364 (78.8)**	**98 (21.2)**
Yes	**1440 (63.2)**	**840 (36.8)**	**1049 (65.4)**	**554 (34.6)**	**391 (57.8)**	**286 (42.2)**
**Number of partners**						
≤5	**1833 (77.6)**	**530 (22.4)**	**1432 (78.3)**	**396 (21.7)**	**401 (75)**	**134 (25)**
>5	**961 (61.6)**	**600 (38.4)**	**607 (63.4)**	**350 (36.6)**	**354 (58.6)**	**250 (41.4)**
**Condomless anal sex^1^**						
No	**975 (78.9)**	**261 (21.1)**	**818 (80.2)**	**202 (19.8)**	157 (72.7)	59 (27.3)
Yes	**1819 (67.7)**	**869 (32.3)**	**1221 (69.2)**	**544 (30.8)**	598 (64.8)	325 (35.2)
**STI**						
No	**2400 (73.0)**	**889 (27.0)**	**1796 (74.4)**	**619 (25.6)**	**604 (69.1)**	**270 (30.9)**
Yes	**394 (62.0)**	**241 (38.0)**	**243 (65.7)**	**127 (34.3)**	**151 (57)**	**114 (43)**
**Steady partner**						
No	1769 (71.5)	704 (28.5)	1342 (73.5)	483 (26.5)	427 (65.9)	221 (34.1)
Yes	1025 (70.6)	426 (29.4)	697 (72.6)	263 (27.4)	328 (66.8)	163 (33.2)
**HIV perceived risk**						
No/Low	**2096 (72.7)**	**786 (27.3)**	**1452 (76.1)**	**457 (23.9)**	644 (66.2)	329 (33.8)
Moderate/High/100% sure	**560 (64.6)**	**307 (35.4)**	**476 (64.8)**	**259 (35.2)**	84 (63.6)	48 (36.4)

All results are shown as number of respondents in that category followed by percentage in parentheses; bold: *p* < 0.05.

**Table 3 ijerph-19-01704-t003:** Factors associated with SDU, overall and stratified by current PrEP use.

			Current PrEP Use			
	All (N = 3924)		No (N = 2785)		Yes (N = 1139)	
	OR (95% CI)	aOR (95% CI)	OR (95%CI)	aOR (95% CI)	OR (95%CI)	aOR (95% CI)
**Current PrEP use**						
No	Ref.	Ref.	NA	NA	NA	NA
Yes	**1.39 (1.20–1.61) *****	**1.19 (1.00–1.41) ***	NA	NA	NA	NA
**Recruitment**						
On site	0.99 (0.84–1.16)	NA	NA	NA	NA	NA
Online	Ref.	NA	NA	NA	NA	NA
**Gender**						
Cisgender men	Ref.	Ref.	Ref.	Ref.	Ref.	Ref.
Transgender women	**1.79 (1.39–2.29) *****	**2.44 (1.75–3.39) *****	**2.09 (1.55–2.81) *****	**2.54 (1.71–3.77) *****	1.25 (0.78–1.97)	**2.11 (1.13–3.92) ***
Non-binary	1.28 (0.81–1.98)	1.52 (0.92–2.46)	1.53 (0.90–2.53)	**1.85 (1.04–3.21) ***	0.82 (0.31–1.92)	0.87 (0.30–2.34)
**Geographical region**						
North/Northeast/Central-west	Ref.	Ref.	Ref.	Ref.	Ref.	Ref.
South/Southeast	1.16 (0.98–1.37)	**1.26 (1.04–1.53) ***	0.99 (0.81–1.21)	1.13 (0.90–1.43)	**1.65 (1.22–2.27) ****	**1.64 (1.16–2.34) ****
**Age (years)**						
18–24	**1.37 (1.11–1.69) ****	**1.41 (1.10–1.81) ****	**1.52 (1.19–1.94) *****	**1.42 (1.06–1.89) ***	1.12 (0.76–1.66)	1.32 (0.82–2.13)
25–35	**1.33 (1.13–1.56) *****	**1.24 (1.04–1.53) ***	**1.28 (1.05–1.56) ***	1.13 (0.91–1.41)	**1.48 (1.12–1.95) ****	**1.54 (1.12–2.13) ****
>35	Ref.	Ref.	Ref.	Ref.	Ref.	Ref.
**Race/color**						
White/Asian	1.05 (0.91–1.21)	**1.21 (1.02–1.42) ***	0.96 (0.81–1.14)	1.21 (0.99–1.48)	1.28 (0.99–1.64)	1.20 (0.89–1.62)
Black/Pardo/Indigenous	Ref.	Ref.	Ref.	Ref.	Ref.	Ref.
**Completed schooling**						
Elementary	Ref.	Ref.	Ref.	Ref.	Ref.	Ref.
Secondary	0.81 (0.60–1.10)	1.00 (0.69–1.46)	0.72 (0.51–1.03)	0.99 (0.65–1.52)	1.10 (0.58–2.15)	1.20 (0.55–2.72)
Post-secondary	0.78 (0.58–1.04)	1.10 (0.75–1.63)	**0.64 (0.47–0.90) ****	1.01 (0.66–1.58)	1.25 (0.68–2.40)	1.49 (0.66–3.50)
**Family monthly income**						
Low	Ref.	Ref.	Ref.	Ref.	Ref.	Ref.
Middle	0.92 (0.78–1.08)	1.06 (0.87–1.30)	0.87 (0.71–1.06)	1.08 (0.85–1.38)	1.02 (0.76–1.37)	1.01 (0.70–1.47)
High	1.09 (0.91–1.31)	**1.32 (1.05–1.67) ***	1.00 (0.80–1.25)	1.29 (0.98–1.71)	1.32 (0.95–1.83)	1.42 (0.91–2.21)
**Binge drinking**						
No	Ref.	Ref.	Ref.	Ref.	Ref.	Ref.
Yes	**2.72 (2.34–3.18) *****	**2.66 (2.25–3.14) *****	**2.72 (2.27–3.28) *****	**2.56 (2.10–3.14) *****	**2.72 (2.08–3.57) *****	**2.85 (2.12–3.86) *****
**Number of partners**						
≤5	Ref.	Ref.	Ref.	Ref.	Ref.	Ref.
>5	**2.16 (1.88–2.49) *****	**1.88 (1.61–2.21) *****	**2.09 (1.76–2.48) *****	**1.86 (1.53–2.25) *****	**2.11 (1.64–2.73) *****	**1.93 (1.45–2.57) *****
**Condomless anal sex**						
No	Ref.	Ref.	Ref.	Ref.	Ref.	Ref.
Yes	**1.78 (1.52–2.09) *****	**1.49 (1.25–1.79) *****	**1.80 (1.50–2.17) *****	**1.63 (1.33–2.01) *****	**1.45 (1.05–2.02) ***	1.18 (0.82–1.71
**Reported STI**						
No	Ref.	Ref.	Ref.	Ref.	Ref.	Ref.
Yes	**1.65 (1.38–1.97) *****	**1.40 (1.14–1.71) ****	**1.52 (1.20–1.91) *****	1.27 (0.97–1.65)	**1.69 (1.27–2.24) *****	**1.53 (1.11–2.09) ****
**Steady partner**						
No	Ref.	NA	NA	NA	NA	NA
Yes	1.04 (0.91–1.20)	NA	NA	NA	NA	NA
**HIV perceived risk**						
No/Low	Ref.	Ref.	Ref.	Ref.	Ref.	Ref.
Moderate/High/100% sure	**1.46 (1.24–1.72) *****	**1.37 (1.14–1.64) ****	**1.73 (1.44–2.08) *****	**1.38 (1.13–1.70) ****	1.12 (0.76–1.63)	1.35 (0.89–2.05)

OR: odds ratio (bivariate analysis); aOR: adjusted odds ratio (multivariate analysis); CI: confidence interval; * *p* < 0.01; ** *p* < 0.001; *** *p* < 0.0001; no relevant collinearity verified by VIF among variables retained in final multivariate models (VIF < 1.84).

## Data Availability

The data presented in this study are available on request from the corresponding author. The data are not publicly available due to important vulnerabilities of the study population that prevent us from unrestricted disclosure of the data.

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
