# Peer review of "High Rates of Sexualized Drug Use or Chemsex among Brazilian Transgender Women and Young Sexual and Gender Minorities"

_ijerph, 2022, doi:10.3390/ijerph19031704_

Round 1
Reviewer 1 Report
The current study reported on the relations among various predictor variables and sexualized drug use (SDU) among Brazilian sexual and gender minorities. A large convenience sample of on-site and online participants were recruited, and results revealed that transgender women had the highest SDU odds when compared to men who have sex with men and non-binary individuals. The potential relation between SDU and HIV vulnerability is noted along with the importance of addressing HIV prevention amongst the gender and sexual minorities that were studied. There are a number of comments that I have and are listed below.
Page 2, line 59. “National data point to …” Should be National data for Brazil…
Page 2, line 60. “One in four TGW …” Should be One in four TGW living in Brazil…
Page 3, line 106. What is the rationale for grouping Asian with White? How many Asians were grouped with White needs to be provided at some point. Also need to know how many Indigenous were grouped with Pardo?
Page 3, line 108. Assessment of Family income is somewhat problematic since we do not know how many people were included in the Family. People living alone versus living with family would have very different “family” incomes. Thus, any results with “Family” income would be suspect.
Page 3, lines 110-113. Substance use is very broad in a yes/no format. Cannot differentiate levels of drug use.
Page 3, line 119. Were the response categories for sexual behavior in binary form (this should be noted in the text). Again, this makes the variables very broad with no room for differentiating levels of sexual behavior.
Page 3, line 121-123. What were the frequencies for the different response categories for HIV perceived risk (although reported in Suppl Table 2 it forces the reader to figure that out)? What was the rationale for creating the binary variable of no/low and moderate/high/100% sure?
Page 3, lines 142-143. “…; our hypothesis is that these associations may differ among individuals on and off-PrEP.” Where is the rationale for this open ended moderation hypothesis?
Page 3, line 144. “…and retained in the final multivariable model.” Exactly what type of multivariable logistically regression was used? Simultaneous entry? Stepwise entry? Forward entry? Backward entry? What was the rationale for the chosen method of entry?
Page 3, Results. Is there a way of comparing the characteristics of the current sample with other samples so that one can get some notion as to whether the current sample is somewhat representative of a larger population as opposed to a selection bias issue? For instance, are the demographic characteristics of the TGW similar to what other researchers have found for TGW in Brazil?
Page 4, lines 149-153. Given that the n’s for some groups are quite large any small differences are likely to show statistical significance it would be helpful to know the effect sizes and to mention them otherwise the reader gets a distorted picture of differences. For Table 1 it would be helpful if the authors include effect sizes.
Page 4, line 154. Besides the mentioned difference between the recruited online vs. onsite (Off-PrEP) were there any other differences across all of the variables between the recruited online vs. onsite participants?
Page 5, line 177-183. It appears that the reported results (aOR) are adjusted odds ratios. They are adjusted for what variables. It would seem that the authors should present the bivariate relations between the predictor variables and the outcome variable (SDU) so that the reader knows the initial relations and then which predictors met the alpha criterion of ≤ .20 (this information is presented in Supplement Table 2 but it might be better to include in the main body as opposed to a supplement). Where do the authors indicate in the text which variables were included in the multivariable logistical regression analysis? Which predictor variables remain statistically significant when included in the multivariable logistical regression analysis? In line with this were the statistical assumptions for the logistical regression assessed (e.g., multicollinearity)?
Page 5, lines 185-193. The authors present their findings for PrEP users regarding SDU predictors and then (and separately) report the findings for off-PREP users. It would seem that the PrEP variable is a moderator variable and thus should be incorporated into the analysis as a moderator variable. The question should be does PrEP (on vs off) moderate the bivariate relations between the predictors and SDU?
Discussion. It is not clear as to whether the reported findings are the bivariate findings or the findings from the multivariable logistical regression analysis.
Discussion-Limitations. The study did not examine whether any of the predictors acted as moderator variables (the study did address the PrEP variable but not as a statistical moderator variable in the logistical regression analysis). Was Covid-19 an issue in Brazil during data collection?
Reviewer 2 Report
Most of the methodological problems are discussed in the study limitations. The authors did not address a potential problem comparing TGW with MSM as in the sample the number of females is so much smaller than the number of males. Was this difference in numbers the same in both recruitment methods? I think that this might be addressed when discussing recruitment of the sample.
This is a well written straight forward presentation . In terms of the English have a problem with the wording which occurs on line 250. What do the authors mean when they say that the risks associated with drinking alcohol have been 'undermined'? I think either a different word or an explanation is called for.
I also found it confusing to have 2 Table 2s. A better numbering system that would distinguish the tables would be helpful
